# Growing Teratoma Syndrome in the Setting of Sarcoidosis: A Case Report and Literature Review

Adel Shahnam [1,*], Robyn Sayer [2], Unine Herbst [3], Raghwa Sharma [4], Won-hee Yoon [1], Tim Dinihan [5] and Bo Gao [1,*]

1 Medical Oncology Department, Blacktown and Westmead Hospitals, Sydney, NSW 2145, Australia; won-hee.yoon@health.nsw.gov.au
2 Gynecological Oncology Department, Chris O'Brien Lifehouse, Sydney, NSW 2050, Australia; robynsayer@hotmail.com
3 Gynaecological Oncology Department, Westmead Hospital, Sydney, NSW 2145, Australia; unine.herbst@health.nsw.gov.au
4 Institute of Clinical Pathology and Medical Research, Westmead Hospital, Sydney, NSW 2145, Australia; raghwa.sharma@health.nsw.gov.au
5 Respiratory and Thoracic Medicine, Blacktown Hospital, Sydney, NSW 2148, Australia; timothy.dinihan@health.nsw.gov.au
* Correspondence: adel.shahnam@health.nsw.gov.au (A.S.); bo.gao@health.nsw.gov.au (B.G.); Tel.: +61-288-905-200 (B.G.)

**Abstract:** Growing teratoma syndrome (GTS) is rare and can mimic disease recurrence in patients with a history of immature teratoma. Benign hypermetabolic lymphadenopathy found on staging and surveillance computed tomography (CT) and positron emission tomography (PET) may lead to the presumption of metastatic malignancy. We report a case of a 38 year old with mixed mature and immature teratomas who developed new peritoneal masses after adjuvant chemotherapy despite a normalization of tumor markers. In addition to low FDG uptake observed in these peritoneal masses, a PET scan showed hypermetabolic lymphadenopathy and pulmonary and spleen lesions suggesting widespread metastases. Subsequent surgical resection confirmed a mixed pathology with GTS and sarcoidosis. We reviewed the current literature evidence of GTS and sarcoidosis as a benign cause of lymphadenopathy in cancer patients. We emphasize the importance of a tissue diagnosis before instituting therapy for presumed cancer recurrence to avoid potentially fatal diagnostic traps and management errors. A multiple disciplinary team approach is imperative in managing patients with suspected recurrent immature teratomas.

**Keywords:** growing teratoma syndrome; immature teratoma; sarcoidosis

## 1. Introduction

Immature teratomas of the ovary represent approximately 1% of ovarian tumors [1]. It is composed of tissues derived from all three embryonic layers: endoderm, mesoderm and ectoderm—with at least one lacking full differentiation. They are graded according to the proportion of immature neuroepithelial tissue. The grade and stage of immature teratomas have prognostic significance and are used to make therapeutic decisions [2]. Tumor markers beta-HCG and alpha-feta-protein (AFP) can be useful in establishing the diagnosis, monitoring treatment response and detecting recurrence [3].

Completion surgery with comprehensive staging is recommended as initial surgery for patients with immature teratoma who do not desire fertility preservation [4]. If patients have had incomplete surgical staging, completion surgery should be considered. Fertility sparing surgery is recommended for those desiring fertility preservation, regardless of stages [5]. After surgery, observation with surveillance is the recommended option for patients with grade 1 stage I immature teratomas. For patients with stage I, grades 2 to 3 or stages II–IV immature teratomas, adjuvant chemotherapy with three to four cycles

of BEP (bleomycin, etoposide and cisplatin) is the standard of care [3]. Patients achieving complete response after chemotherapy should be followed up every three months (AFP, beta-HCG levels) for two years then six monthly [6]. Radiographic imaging with CT is also recommended as part of surveillance. In the first year, CT should be performed every three months and the frequency reduced to six monthly in the second year, and then annually after three years and up to five years of follow-up post treatment [6].

Most patients with immature teratomas are cured, but a small percentage develop disease recurrence. It has been reported that 90% of recurrence occurs within 24 months of the initial diagnosis [7]. Because of the rarity of relapse in this population, the treatment strategies are extrapolated from clinical experience with testicular cancer. The most commonly used salvage regimens include TIP (paclitaxel, ifosfamide, cisplatin), VIP (etoposide, ifosfamide, cisplatin) or VeIP (vinblastine, ifosfamide, cisplatin) [8]. For patients with platinum-refractory disease, defined as progression within four weeks of completion of platinum-based chemotherapy, the prognosis is poor and high-dose chemotherapy with stem-cell rescue is often recommended [9].

## 2. Case Report/Case Presentation

A 38-year-old female presented to a local hospital with acute abdominal pain in early 2018. She had a history of a left dermoid cyst. Physical examination found a 10 cm tender left-lower abdominal mass. She had an elevated AFP of 160 IU/mL and an undetectable beta-HCG. The provisional diagnosis was a ruptured dermoid cyst and an urgent laparoscopic left salpingo-oophorectomy was performed. Pathology revealed a grade 3 immature teratoma. Her subsequent staging CT scan showed several tiny pulmonary nodules measuring 2 to 3 mm and multiple small ill-defined hypodensities within the splenic parenchyma of unknown etiology. In addition, there was a borderline enlarged pre-carinal lymph node measuring 17 by 12 mm.

A completion surgery with comprehensive staging was performed in a tertiary referral hospital. Scattered deposits/implants were observed on the surface of the abdominal/pelvic cavity, which was completely resected macroscopically. The final pathology revealed a staged 3B mixed grade 3 immature teratoma (Figure 1A) and mature teratoma (Figure 1B).

She received four cycles of BEP adjuvant chemotherapy and tumor markers (AFP) normalized during chemotherapy. Repeat CT at the completion of chemotherapy however demonstrated new peritoneal masses (Figure 2A). The bilateral pulmonary nodules and hypodense splenic lesions remained unchanged. A fluorine-18 fluorodeoxyglucose (PDG) PET scan was performed to further characterize these findings. It demonstrated mild FDG uptake (SUVmax of 3.9) in the peritoneal masses (Figure 2B). Furthermore, there were intensely metabolic avid (SUVmax up to 13.3) mediastinal, porta hepatis/peripancreatic lymphadenopathy (Figure 2C) and spleen lesions (Figure 2D). An endobronchial ultrasound (EBUS) with fine-needle aspirate cytology (FNAC) of station 7 and 4 lymph nodes showed highly atypical small cells. In the presence of cartilage, these could be suspicious of metastatic teratomas. In addition, there was evidence of granulomatous inflammation.

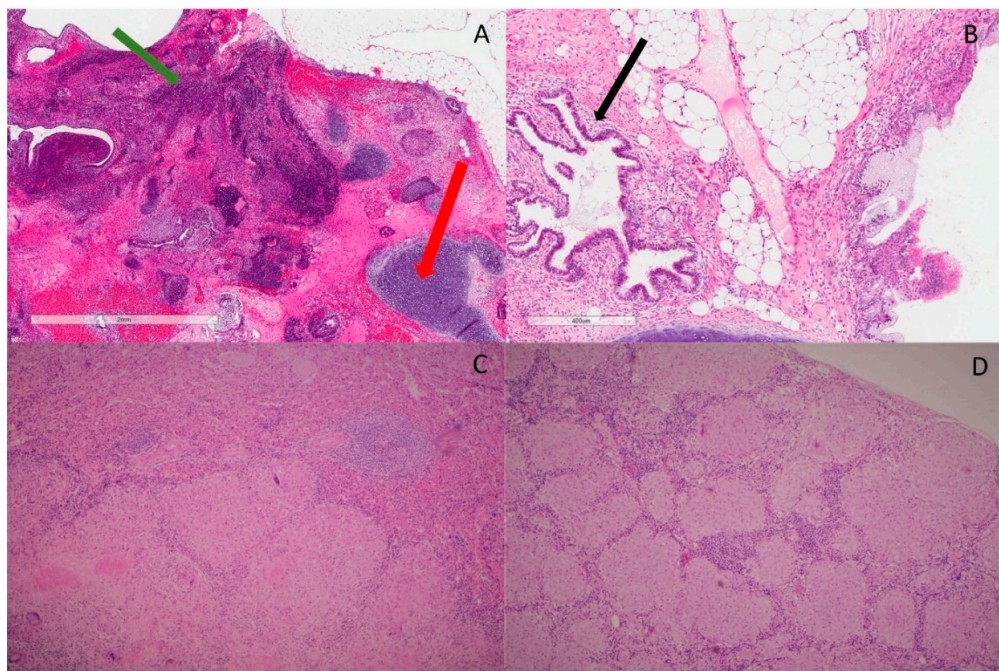

**Figure 1.** Histopathological features. (**A**) H&E slide ×20 magnification shows area of immature teratoma comprising primitive neuroectodermal elements (green arrow). There is surrounding cartilage, which represents part of mature teratoma (red arrow); (**B**) H&E slide ×20 magnification shows benign glands' architecture (black arrow) with fibroblasts, fat and a small portion of cartilage in the background. No immature teratoma is identified; (**C**) H&E slide ×40 magnification shows large granulomas with epithelioid and Langerhans's giant cells, focally with central hyalinization and necrosis in sections of spleen. Mycobacterial and fungal stains are negative; (**D**) H&E slide ×40 magnification shows well-formed, noncaseating granuloma with few surrounding lymphocytes in porta hepatis lymph nodes.

Her imaging and pathology were reviewed in our Gynecological Oncology multidisciplinary team (MDT) meeting. The peritoneal masses with low FDG uptake were consistent with growth teratoma syndrome (GTS). The hypermetabolic lymphadenopathy and splenic lesions could have been due to metastatic immature teratomas, but other benign conditions, such as infectious etiologies or inflammatory diseases, could not be ruled out. A careful review of FNAC showed scattered group of small, atypical cells that were devoid of mitoses and marked nuclear atypia. Therefore, the differential diagnoses included benign endobronchial cells with cartilage from respiratory tract origin or components of recurrent teratomas. The management options of surgical resection or second-line high-dose chemotherapy with stem cell rescue were debated. The final consensus was to proceed with a laparotomy to completely resect the mildly FDG avid peritoneal masses and highly FDG avid porta hepatis lymph nodes and splenic lesions.

In July 2018, the patient had the third procedure, including a laparotomy, tumor debulking and splenectomy. The final pathology confirmed GTS involving the peritoneal lesions. However, the spleen, porta hepatis nodes, vaginal vault and liver surface nodules showed large active non-caseating granulomas with epithelioid and Langerhans giant cells with focal central hyalinization and necrosis (Figure 1C,D) consistent with sarcoidosis. The lesions in the mediastinum and chest had a similar FDG uptake as the sarcoidosis in the spleen and portal hepatis, suggesting her chest disease was likely sarcoid rather than a growing teratoma or metastatic immature teratoma. This patient is now undergoing regular follow-up and remains disease-free 3.5 years after the third surgery.

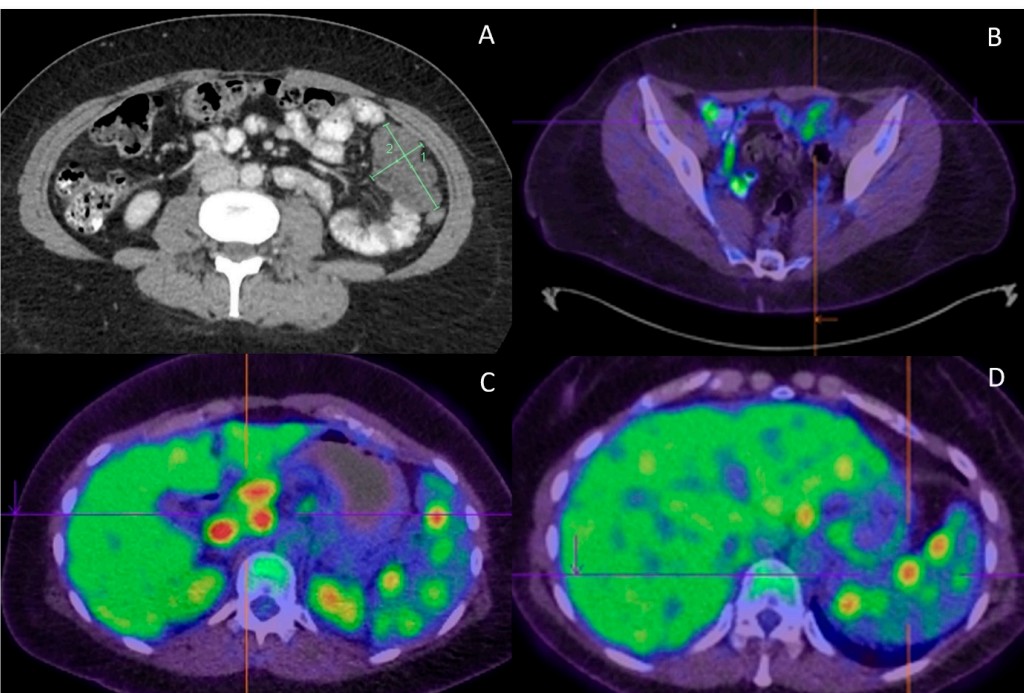

**Figure 2.** Radiological features. (**A**) Progress CT at completion of adjuvant chemotherapy found new peritoneal lesions; (**B**) Peritoneal lesions demonstrate only low FDG uptake (SUVmax of 3.9); (**C**) On the contrary, portal hepatis/peripancreatic lymphadenopathy is highly FDG avid (SUVmax of 13.3); (**D**) High FDG uptake (SUVmax of 9.3) is also observed in the splenic lesions.

## 3. Discussion

In this report, we described a female with resected mixed mature and immature teratomas who had a complete biochemical response to adjuvant BEP chemotherapy, but developed a diagnostic challenge of enlarging peritoneal lesions. Staging PET scan showed hypermetabolic lymphadenopathy and pulmonary nodules and splenic lesions. She was subsequently found to have dual pathology with GTS and sarcoidosis.

GTS was first reported by Logothetis in 1982, after observing enlarging solitary mass(es) following chemotherapy, despite normal tumor markers in six patients with metastatic mixed germ-cell tumors [10]. It was confirmed by the findings of mature teratomas with an absence of malignant histologies following tumor resection. Because of the chemo-resistant nature of mature teratomas, it is imperative to recognize GTS to prevent further ineffective systemic treatment. Early surgical resection is currently the gold standard treatment for GTS, preventing growth into critical organs [11,12].

The incidence of GTS in males with non-seminomatous germ-cell tumors has been estimated at around 2–8%. Two recent, large case series reported GTS to occur in 20% of patients with immature teratomas and half of GTS occurred during chemotherapy [13–15]. This underscores the importance of routine surveillance with imaging the patients with immature teratomas. GTS should be considered in the differential diagnosis, if new lesions are observed.

The pathogenesis of GTS is still uncertain. Two different mechanisms have been proposed. The first is that chemotherapy induces the malignant cell differentiation of an immature teratoma into a mature teratoma. The second is that chemotherapy selectively destruct immature components that are chemotherapy sensitive, whereas mature components, resistant to chemotherapy, persist and grow as GTS [11]. GTS does have a potential risk of malignant transformation with reports of transformation into sarcomas, adenocarcinomas and other rare malignancies occurring in 5% of patients [15].

The risk factors for GTS have not been confirmed, but the presence of residual disease at primary surgery is considered an important contributor [16]. GTS generally develops

within two years of initial diagnosis, however there have been reports where recurrences occurred beyond five years after initial treatment [17]. The retroperitoneum appeared in the most common site of disease, however other sites, such as lung, cervical, supraclavicular, inguinal lymph nodes, mediastinum, forearm, mesentery and liver, have been described [17].

In patients with immature teratomas, GTS can be mistaken for disease relapse or recurrence. The diagnosis of GTS can be challenging, but should be suspected after serial imaging showing enlarging mass(es) during or after treatment with chemotherapy with discordant reduction in tumor markers [12]. There are several radiological features on CT that can assist in identifying a lesion as GTS. These features include margins that are well circumscribed, high Hounsfield units, cystic variations, internal calcification and fatty areas [18].

FDG-PET/CT is widely used for cancer staging and surveillance. FDG-avid lymph nodes, however, are not specific for malignancy [19]. Inflammatory cells avidly take up FDG when undergoing the energy-dependent process of activation. Benign disorders as differential diagnosis of diffuse lymphadenopathy include (1) infectious etiologies, such as histoplasmosis and HIV; (2) inframammary disease due to granulomatous and autoimmune diseases; and (3) benign lymphoproliferative disease, such as Castleman's disease [20]. The interpretation of this patient's PET scan was impacted by the undiagnosed systemic sarcoidosis, a disease that often demonstrates avidity on PET [21].

Sarcoidosis is a multisystemic, non-caseous, benign granulomatous disease of unknown etiology with the incidence estimated to be 50 to 150 per 100,000 population [22]. The diagnosis of sarcoidosis is based on the histopathological finding demonstrating non-caseating granulomas [21]. Sarcoidosis can affect any organ, including the liver and spleen. Characteristic CT imaging findings include bilateral hilar and mediastinal lymphadenopathy, as well as non-specific parenchymal changes of nodules [23]. Patients with sarcoidosis are often asymptomatic, as was the case for our patient [24]. The symptoms of sarcoid are dependent on the organ(s) involved, but the most common include cough, dyspnea, chest pain and fevers [25]. Sarcoidosis can have high avidity with FDG-PET [26]. The mainstay treatment for symptomatic sarcoidosis involves a glucocorticoid steroid with complete response observed in one-third of patients and a partial response in another one-third of patients [27]. Our patient was asymptomatic for sarcoidosis, and was therefore managed with surveillance.

Several small case reports have described the coexistence of sarcoidosis and malignancy, including Hodgkin's lymphoma and testicular cancer. In a retrospective case review of 565 patients who underwent mediastinoscopy at a Swedish Medical Center and Cancer Institute, the prevalence of biopsy-proven sarcoidosis was 7.5% [28]. Even accounting for referral bias, this was significantly higher than the estimated 50 to 150 per 100,000 of the general population. No definite causal relationship was identified. It is unclear whether the chronic inflammation and possible decreased immune surveillance associated with sarcoidosis may predispose patients to develop cancer or, conversely, the host immune response to cancer or cancer therapy may predispose to sarcoidosis [29,30]. Certain chemotherapeutics, particularly alpha-interferon and bleomycin, which was received by this patient, were reported to predispose to the development or activation of sarcoidosis [31,32]. There are emerging case reports of sarcoidosis with immunotherapy in cancer patients [33]. Consequently, due to these associations, core biopsies should be the gold standard to confirm a malignant diagnosis.

## 4. Conclusions

In conclusion, GTS is a rare entity that can cause diagnostic uncertainty, post-treatment of immature teratomas. A multiple disciplinary team approach is imperative in managing patients with suspected recurrent immature teratomas. Hypermetabolic lymphadenopathy on staging or surveillance imaging can present a diagnostic dilemma. To avoid potentially fatal diagnostic traps and management errors, it is important to obtain adequate tissue

through a core needle biopsy whenever possible, or even a surgical biopsy if necessary, before instituting therapy for presumed cancer recurrence.

**Author Contributions:** A.S.: manuscript writing, case review, literature review; R.S. (Robyn Sayer): manuscript writing, case review, literature review; U.H.: manuscript writing, case review, literature review; R.S. (Raghwa Sharma): manuscript writing, case review, literature review; W.-h.Y.: manuscript writing, case review, literature review; T.D.: manuscript writing, case review, literature review; B.G.: manuscript writing, case review, literature review, supervisor. All authors have read and agreed to the published version of the manuscript.

**Funding:** No funding was obtained.

**Institutional Review Board Statement:** The study was conducted in accordance with the Declaration of Helsinki, and approved by The Western Sydney Local Health District Human Research Ethics Committee (protocol code 2111-13 CR on 25 November 2021).

**Informed Consent Statement:** Written informed consent has been obtained from the patient(s) to publish this paper.

**Data Availability Statement:** All information from this case is stored in our center's electronic health records.

**Conflicts of Interest:** The authors have no conflicts of interest to declare.

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
