# Peer review of "Growing Teratoma Syndrome in the Setting of Sarcoidosis: A Case Report and Literature Review"

_curroncol, doi:10.3390/curroncol29060331_

Round 1
Reviewer 1 Report
Interesting case report and review of the literature. Not innovative.
minor typos: i.e. line28 mesodermal, line 32 APF, line 61 sampingo, line 189 multiple disciplinary
Reviewer 2 Report
Thank you for this interesting case report. It emphasizes very well the need for multidisciplinary tumor boards by characterizing the risk of over- and under-treatment.
A few comments though:
Spelling: Line 132 deferential => differential
Line 154 PDG => FDG
Discussion:
Line 175-188: Sarcoidosis or Sarcoid like lesions represent a known challange in patients with cancer. You might want to add, that a core needle biopsie should be the gold standard to secure a malignant diagnosis. See case series: Eggers et al. Oncol Res Treat 2019;42:382–386
General remarks:
Why was third surgery performed extensively including splenectomy? Did you discuss further biopsy of low and high FDG-avid leasion before?
Following these thoughts, you might be able to be more specific to define "adequate tissue diagnostic" in you conclusion. I would suggest that a biopsy should be preferred whenever possible to prevent overtreatment.
